# Applications of Nano/Micromotors for Treatment and Diagnosis in Biological Lumens

**DOI:** 10.3390/mi13101780

**Published:** 2022-10-19

**Authors:** Shandeng Huang, Yinghua Gao, Yu Lv, Yun Wang, Yinghao Cao, Weisong Zhao, Dongqing Zuo, Haoran Mu, Yingqi Hua

**Affiliations:** 1Department of Orthopedics, Shanghai General Hospital, School of Medicine, Shanghai Jiao Tong University, Shanghai 200080, China; 2Shanghai Bone Tumor Institution, Shanghai 201620, China

**Keywords:** nano/micromotor, biological lumen, locomotion

## Abstract

Natural biological lumens in the human body, such as blood vessels and the gastrointestinal tract, are important to the delivery of materials. Depending on the anatomic features of these biological lumens, the invention of nano/micromotors could automatically locomote targeted sites for disease treatment and diagnosis. These nano/micromotors are designed to utilize chemical, physical, or even hybrid power in self-propulsion or propulsion by external forces. In this review, the research progress of nano/micromotors is summarized with regard to treatment and diagnosis in different biological lumens. Challenges to the development of nano/micromotors more suitable for specific biological lumens are discussed, and the overlooked biological lumens are indicated for further studies.

## 1. Introduction

Biological lumens in the human body are the main natural systems for the delivery of nutrients, metabolites, and even cell activities. The invention of manual delivery systems for disease treatment and diagnosis partly depends on the anatomic features of biological lumens, such as the digestive tract, blood vessels, ureter, bronchi, vagina, etc. [1]. Challenges limiting the development of manual delivery systems include the need to overcome the hydromechanics of the contents in biological lumens, such as gastrointestinal peristalsis, respiration, urination, and even the bloodstream [2,3]. Although possessing enhanced permeability and retention (EPR) and the ability to accumulate at specific targets in sites of inflammation, cancer, or other specific sites, non-motivated delivery systems still drift along in biological lumens for hundreds and thousands of cycles before being removed from the body. However, nano/micromotors offer an approach to solving these problems by manual control or pre-set procedure [4]. Nano/micromotors could perform locomotion and overcome hydromechanics. By providing an external force or a starting mode, these motors resist the motion in the biological lumens or even perform a reverse motion, showing great potential for delivery systems for treatment application.

Nano/micromotors perform locomotion by utilizing different chemical or physical power sources [5], such as chemical fuel [6,7,8,9,10,11], optical [12,13], ultrasonic [14,15,16], magnetic [17,18,19,20,21,22,23,24] and external thermal [25]. Self-powering motors or those driven by external equipment move regularly, predictably, and controllably rather than by Brownian movement. This locomotion could also be achieved by utilizing or mimicking the microorganisms or body cells, which can crawl and wriggle [26,27,28,29]. Power sources specifically applicable to some biological lumens are not suitable for other lumens. For example, motors driven by chemical fuel might not be suitable for blood application because the off-gas of the motors might embolize the small vessels in the lung or brain. Additionally, the external power reduces as the lumens extend deeper into the tissue, making it difficult for nano/micromotors to travel further.

In this review, we summarize the research progress of nano/micromotors with regard to delivery systems in different biological lumens (Appendix A). However, some challenges remain to be overcome, among which propulsion, control, and biocompatibility are the main problems. It is difficult for micromotors to propel on the micro-scale where the Reynolds number is very low. Due to the location of most lumens deep within the body, controlling the swarm of micromotors may seem difficult to achieve. To overcome these, the micromotors with chemical fuel have a strong driving force but a poor lifespan. External fields give micromotors controllability, meanwhile reducing the velocity. Cell-like micromotors combine the efficient movement of synthetic micromotors and the biological functions of natural cells, showing full biocompatibility. Together, we give an outlook on the challenges in the development of nano/micromotors suitable for specific biological lumen and indicate the overlooked biological lumens in the human body for delivery system establishment.

## 2. Nano/Micromotors in the Gastrointestinal Tract

The gastrointestinal tract has slow motility, and it has no interference, such as fluid or food, during the non-feeding periods. Even after food intake, gastrointestinal peristalsis is slow and rhythmic. To anchor in and penetrate the gastrointestinal wall, drugs must overcome many barriers, including enzymatic, sulfhydryl, mucus, and epithelial barriers [30]. The strong acid environment in the stomach, astronomical small intestinal folds, and abundant microorganisms in the large intestine hamper drug delivery.

The special acidic environment in the stomach provides fuel for the redox reactions of active metals such as zinc (Zn) and magnesium (Mg). Satayasamitsathit et al. [31] demonstrated nano/micromotors based on Zn, fabricated by electrodeposition (Table 1). This nano/micromotor had a biconical structure and could move rapidly in an acidic environment. The directional locomotion was enabled by the formation of a galvanic cell between the Zn and the sputtered gold contact. The different sizes of the payload were mixed with the Zn, providing an extremely high theoretical drug-loading rate of 74%. The drugs were released along with the propulsion, resulting in the loss and dilution of drugs. Gao et al. [32] reported similar nano/micromotors based on Zn capable of moving inside a mouse’s stomach (Table 1). The nano/micromotor had the biconical structure coated with poly (3,4-ethylene dioxythiophene) (PEDOT), using a membrane template process. Compared with the previous nano/micromotor, it greatly extended the life of the nano/micromotor in the stomach to 10 min and improved the penetration and retention of nano/micromotors in the stomach through the double-layer structure, which reduced the velocity. De Avila et al. [33] demonstrated a multi-chamber biconical tubular nano/micromotor, which showed strong propulsion in the acidic gastric environment, thereby facilitating its distribution in the stomach and penetration into the gastric mucosal layer (Figure 1a, Table 1). The structure of this nano/micromotor was divided into two parts. The first part was the propulsion element based on zinc; the second part was the drug delivery element. When the nano/micromotor penetrated the mucosal layer, the pondus hydrogen(pH)-responsive cap of the drug delivery element dissolved and released the loaded drugs, greatly reducing the loss of drugs during the transportation and propulsion process. This nano/micromotor achieved efficient propulsion, effective drug delivery, good biocompatibility, and four-fold enhanced retention time compared to the mono-compartment motors in mouse models.

However, a biconical tubular structure cannot be made into a smaller structure, and an asymmetric structure, called Janus, has been developed to allow nanoscale motors to move. Typical Janus nano/micromotors have the advantages of low cost, simplification, and versatility in drug delivery [39,40]. De Avila et al. [34] designed a magnesium (Mg)-based multi-layer sphere-like Janus nano/micromotor (Figure 1b, Table 1). The Janus nano/micromotor uses the poly (lactic-co-glycolic acid) (PLGA) layer and chitosan layer of the drug clarithromycin (CLR). The model uses stomach acid as fuel to actuate and treat gastritis. When the Mg cores in the motors dissolve, it causes the degradation of the nano/micromotors and the release of drugs. At the same time, the positively charged chitosan outer coating could make the motors adhere to the stomach wall, promoting the effective local release of the drug in PLGA. The researchers optimized its velocity (120μm/s) and life span (about 6 min). However, in the mouse model, there is no statistical significance between micromotors and the free CLR + PPI (proton pump inhibitor) groups. Lin et al. [35] reported a Janus nano/micromotor composed of a Zn core and a gallium (Ga) shell, similar in structure to the Mg-based nano/micromotor designed by de Avila’s group, as described above (Table 1). This nano/micromotor was used as an active-targeted therapeutic platform for antibacterial chemotherapy. The motion of the Ga/Zn nano/micromotor was powered by hydrogen bubbles generated by the spontaneous reaction of the Zn core with gastrointestinal acid. The velocity was enhanced by the Ga-Zn galvanic effect. When the pH value in the simulated gastric juice was 0.5, the velocity could reach 382.3 μm/s, and the life span was about 30 s. With the gradual increase in the pH value, the speed slowed down and the life span increased accordingly. The Ga/Zn nano/micromotor completely degraded in the acidic environment of the gastrointestinal tract and released Ga^3+^, which had high antibacterial activity. The movement of the nano/micromotor improved the diffusion of Ga^3+^ and multiplied the bactericidal effect on Helicobacter pylori by four times compared with the passive Ga^3+^ particles in the stomach, laying the foundation for targeted antimicrobial therapy in the gastrointestinal tract. A common way for micromotors to move is in gastric juice. From another perspective, moving through gastric mucin gels could be an alternative mode. D Walker et al. [41] demonstrated a system of reactive magnetic micromotors that mimic the bacterium *Helicobacter pylori* to move through gastric mucin gels. This was proved in vitro experiments, and is thus a finding of significant medical importance.

Nano/micromotors could change the method of administration. Vaccination is one of the effective methods of preventing infectious diseases, but most of them are administered by intramuscular injection. Wei et al. [36] developed a biomimetic self-propelled micromotion formulation for use as an oral antiviral vaccine (Table 1). A multi-layer spherical Janus nano/micromotor was designed using a magnesium-based core as the power source; the core was covered with red blood cell membranes loaded with bacterial toxins, and was finally covered with a layer of pH corresponding type enteric polymer (Eudragit L100-55, a pH-responsive methacrylate-based polymer). These design innovations allowed natural toxins to pass through the stomach more safely and reach the lower digestive tract, greatly improving the retention and uptake of antigenic substances in the small intestines of mice, and enhancing the antitoxin IgA titer production by approximately 1 order of magnitude compared to the blank solution. Moreover, many bionic nano/micromotors could overcome extrinsic forces. Cai et al. [37] designed a magnesium-based hydrogel nano/micromotor inspired by the sucking disc of an octopus (Figure 1c, Table 1) In the acidic environment of gastric juice, Mg reacted with hydrogen ions to form hydrogen to drive the nano/micromotor, and due to the special sucking-disk-like structure, it could be effectively adsorbed on the gastric mucosa to achieve efficient drug delivery. Even after rinsing 10 times, the micromotors still adhered to the stomach wall tissue ex vivo. Further, in a mouse model of gastric ulcers, the experimental results showed that the drug-loaded nano/micromotor had a better curative effect on the gastric ulcer.
Figure 1Nano/micromotors in the gastrointestinal tract. (**a**) The multi-chamber biconical tubular micromotor is divided into two parts. The first part was the propulsion element based on zinc; the second part is the drug delivery element. When the motor penetrated the mucosal layer, the pH-responsive cap dissolved and released the drugs. Reprinted with permission from ref. [33]. (**b**) The magnesium (Mg)-based multi-layer spherical-like Janus nano/micromotor uses stomach acid as fuel to actuate. Reprinted with permission from ref. [34]. (**c**) The magnesium-based hydrogel nano/micromotor is inspired by the sucking disc of an octopus. Mg reacts with hydrogen ions to form hydrogen to drive the nano/micromotor. Reprinted with permission from ref. [37].
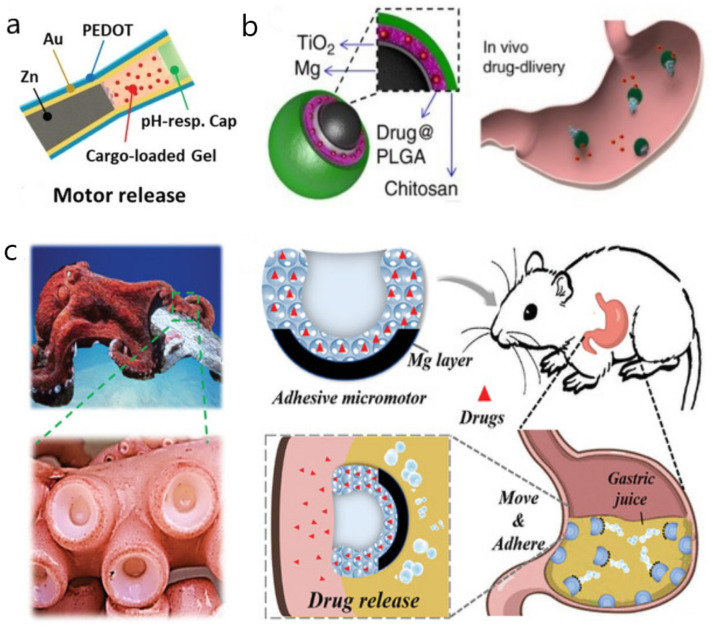


Nano/micromotors with active metals such as Mg and Zn as cores have a limited lifespan, losing their self-propelling property when the metal or fuel runs out. Therefore, nano/micromotors with other driving forces, such as enzymatic reaction and physical fields, need to be explored. Das et al. [15] designed a magnesium-based nano/micromotor with three kinds of propulsion for efficient propulsion in simulated gastric fluids of varying pH and conductivity. The electric-field mode and the chemical propulsion mode worked in very low or high concentrations of simulated gastric juice, respectively, while magnetic rolling was used at intermediate concentrations, effectively bridging the first two propulsion modes.

So far, we have discussed the application of nano/micromotors as a delivery system in the stomach. The gastrointestinal tract includes not only the stomach but also the esophagus, small intestine, colorectum, etc. [42]. The human colorectal mucus layer has a denser and more viscous structure than the stomach, making it difficult for drugs to penetrate and stay [43]. In previous studies, nano/micromotors were mainly used in the large intestine to cure colorectal cancer (CRC), the third most common malignancy and the second most deadly cancer [44]. Chemotherapy by systemic drug administration in the treatment of CRC may cause systemic toxicity, unsatisfactory curative effects, and low tumor-specific selectivity. Targeted therapy based on various monoclonal antibodies also has certain limitations [45]. Oral administration using nano/micromotors would be a viable solution to compensate for the shortcomings of chemotherapy and targeted therapy. Wang et al. [38] fabricated a biological chemotaxis-guided self-thermophoretic nano/micromotor coated with bacteria membrane to improve the efficiency of oral administration in CRC (Table 1). This nano/micromotor facilitated precise intestinal localization and autonomous mucus penetration. The nano/micromotor introduced asymmetric platinum-sprayed mesoporous silica, which was driven by near-infrared ray (NIR) irradiation, to achieve autonomous locomotion to the primary tumor site in the intestine. Additionally, this nano/micromotor was camouflaged using the Staphylococcus aureus membrane to anchor precisely in the CRC-associated gut to protect the nano/micromotor, due to the strong interaction between pathogenic microorganisms and CRC [46,47]. Compared with conventional administration, the nano/micromotor induced a 4.3-fold improvement in biological chemotactic anchoring in the CRC-associated intestine, and a 14.6-fold improvement in autonomous mucus penetration performance, resulting in tumor inhibition in primary CRC mouse xenografts.

Current problems and remaining studies: (A) The effects on the community composition of the microbiota have usually been overlooked in evaluating the toxicity of the micromotors. We suggest that future works should evaluate the effect of micromotors on intestinal microbiota using techniques such as 16S rRNA sequences. (B) The micromotors designed by Wang et al. [38] can identify specific colon cancer segments. Future work may choose some special markers to extend the application of the recognition function to other gastrointestinal tumors or diseases, such as Crohn’s disease. (C) The powering of micromotors in the gastrointestinal tract usually occurs through chemical reaction, which limits the lifespan and propulsion speed of micromotors. We suggest that future works should find alternatives to reactive metals, such as metal hydride, for longer life and better propulsion.

## 3. Nano/Micromotors in the Urinary Tract

The urinary tract is also an important lumen in the body, and is composed of the ureter, bladder, urethra, etc. [48]. Intravesical therapy has been extensively studied for various bladder diseases, such as bladder cancer, interstitial cystitis, and overactive bladder [49,50,51]. However, conventional drug carriers have lower therapeutic delivery efficiency in the urinary tract due to the passive diffusion of drug molecules in the bladder and rapid clearance through periodic urination. The urinary system removes metabolic waste from the body through urination [52]. A large proportion of urea is included among the metabolic wastes in urine. Previous studies have demonstrated that the nano/micromotors could be driven by enzyme catalysis using urea as a substrate and would have good biocompatibility [53]. Ma et al. [54] designed a urea-driven Janus hollow mesoporous silica particle (JHP) nano/micromotor with a lifespan of more than 10 min, and the velocity was positively correlated with the urea concentration in the solution (Figure 2a, (Table 2)). In in vitro experiments, the speed of this nano/micromotor could be regulated by controlling the activity of the enzyme, and the direction of the movement could be remotely controlled with the help of a magnetic field. Choi et al. [55] reported a biocompatible and bioavailable enzyme-powered polymer nano/micromotor, which was fabricated by polydopamine (PDA) hollow nanoparticles (Table 2). The PDA coating could be used to improve cell adhesion and proliferation [56]. This nano/micromotor could penetrate deeply into the mucosal layer of the bladder wall at the depth of 60μm and extend the retention even after repeated urination. Biohybrid nano/micromotors can also be used in the urinary tract. Tang et al. [57] reported an endogenous enzyme-driven Janus platelet nano/micromotor (JPL-motor) system constituted by asymmetrically immobilizing urease on the surface of native platelets (PLTs) (Figure 2b, Table 2). Cell surface engineering using urease had negligible effects on the functional surface proteins of PLTs, and thus, the JPL motors preserved the intrinsic biological functions of PLTs, including efficient targeting of cancer cells [58] and bacteria [59]. The efficient propulsion of JPL motors in the presence of urea fuel greatly enhanced their binding efficiency to these biological targets and their therapeutic efficacy when loaded with model anticancer or antibiotic drugs. In vitro, the anticancer activity of the MDA-MB-231 breast cancer cells of JPL-DOX was 1.75-fold higher than PL-DOX, and the antibacterial efficacy of JPL-Ciprofloxacin (Cip) was 3.6- and 2.1-fold higher than that of Cip and PL-Cip.

Nano/micromotors could also be used in the treatment of urinary tract infections caused by pathogens including Escherichia coli, Klebsiella pneumonia, Proteus mirabilis, and Streptococcus [62,63]. Bacteria begin to infect the urethra from the periurethral area and eventually colonize the bladder and form a bacterial biofilm [63]. Nano/micromotors are widely used to eliminate biofilms in the body. Among them, the treatment methods based on photocatalysis have good therapeutic prospects [64,65]. TiO_2_ is a common photocatalytic material, but TiO_2_ either requires ultraviolet light activation or requires the use of toxic fuels such as H_2_O_2_, which has poor biocompatibility [66]. This limits its medical application. Villa et al. [60] designed an enzyme-photocatalyst tandem nano/micromotor. It could be propelled at a speed of 3.3 ± 0.3 μm/s in 50 mM urea (Figure 2c, Table 2). In vitro experiments, when exposed to visible light, the nano/micromotor was triggered to produce reactive oxygen species (ROS), which could remove nearly 90% of bacterial biofilms, improving the therapeutic effect of urinary tract infections with negligible short-term toxicity. Wu et al. [61] developed a photothermal interference (PTI) urease-modified PDA nano/micromotor (PDA@HSA@Ur) (Figure 2d, Table 2). When exposed to NIR laser light, the nano/micromotors increased the tumor microenvironment temperature. This not only induced tumor cell apoptosis, but also enhanced the biocatalytic activity of urease and the motion of the motor. Compared with nano/micromotors propelled only by urea, the diffusion coefficient of PTI nano/micromotors at 10 mM urea increased significantly to 1.24 µm^2^ s^−1^ in the presence of a 2 W cm^−2^ NIR laser. Furthermore, in vitro, PTI motors showed enhanced anticancer efficiency in HepG2 cells (20% cell viability with laser < 40% without) due to synergistic photothermal and chemotherapeutic effects.
Figure 2Nano/micromotors in the urinary tract. (**a**) The speed and direction of the urea-driven Janus hollow mesoporous silica particle (JHP) nano/micromotor could be regulated. Reprinted with permission from ref. [54]. Copyright© 2022, American Chemical Society. (**b**) The endogenous enzyme-driven Janus platelet nano/micromotor (JPL-motor) turned the urea into NH3 and CO_2_, delivering the DOX efficiently. Reprinted with permission from ref. [57]. (**c**) The enzyme-photocatalyst tandem nano/micromotor could be propelled at a speed of 3.3 ± 0.3 μm/s in 50 mM urea and produced ROS to remove the bacterial biofilms. Reprinted with permission from ref. [60]. (**d**) The velocity of photothermal interference (PTI) urease-modified PDA nano/micromotor (PDA@HSA@Ur) driven by enzyme catalysis could be enhanced when exposed to NIR laser light. Reprinted with permission from ref. [61].
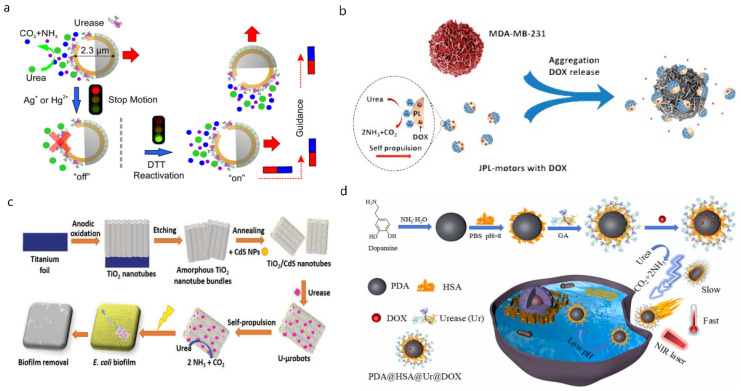


Current problems and remaining studies: (A) Although Villa et al. [60] and Wu et al. [61] developed hybrid light-driven micromotors, the velocity of nano/micromotors in the urinary tract is still much lower than in the gastrointestinal tract. We suggest that future works should find more suitable hybrid drive powering methods, such as electric power, to enable micromotors reach higher speeds. In that case, it is possible to use the navigational technology of the interdigitated microelectrodes (IDE) system to control electric-powered micromotors in vivo. (B) The bladder is not constantly filled with urine. We suggest that future works could design a structure, such as nano-sized Cu-BTC (BTC: benzene-1,3,5-tricarboxylate) [67], that allows the micromotor to store a portion of urea when it is abundant. When the bladder is empty, the micromotor reuses the stored urea, allowing it to move during the bladder’s “dry season”. (C) The lack of penetration and retention of micromotors in the bladder has always been the focus of researchers. However, the excessive penetration and retention ability of the micromotor may lead to the slow clearance of the micromotor. The appropriate penetration and retention capability needs to be evaluated in the future.

## 4. Nano/Micromotors in Blood Vessels

Blood vessels can be found in almost every corner of the human body. The vascular system maintains cellular homeostasis through a complex network of arteries, capillaries, and veins [68]. Administration into blood vessels is mostly accomplished by intravenous injection, which enables the drug to be passively transported to the target site with the flow of blood. Toxicity and side effects cannot be ignored when the drugs take effect. A nano/micromotor delivery system could reduce toxicity and side effects and improve therapeutic efficiency in the blood vessels.

Motors with cores of reactive metals are not suited to the blood environment. The small quantity of fuel in the blood is not enough to propel a nano/micromotor, and the bubbles it makes may lead to the consequence of air embolism. Natural carriers, such as the existing blood cells, might be suitable for the invention of nano/micromotors. Wu et al. [69] applied ultrasound fields to propel Fe_3_O_4_ nanoparticle-loaded RBCs, also called RBC motors (Figure 3a, Table 3). In vitro, RBCs combined with external propulsion systems were successfully controlled even against bloodstream current, making them very interesting for diagnosis and therapy in vascular networks. Gao et al. [70] developed a magnetically navigable red blood cell-mimicking (RBCM) nano/micromotor driven by acoustic energy (Table 3). The nano/micromotor was composed of a double-concave erythrocyte-type hemoglobin core, coated with natural erythrocyte membrane shells, and the interior was encapsulated with photosensitizers (PSs). The oxygen it carried could enhance the killing power of photodynamic therapy on cancer cells. In vitro, over 75% of HeLa cells were killed by the dual-oxygen-indocyanine green (ICG, a kind of PS)-loaded RBCM micromotors, while bare ICG-loaded micromotors had little effect on HeLa cells. The natural red blood cell membrane shell could prevent immune clearance and prolong the circulation time in the body. The nano/micromotor was driven by ultrasound, and the direction of movement was regulated by an external magnetic field, which could move autonomously in vitro. PLTs are also one of the most popular targets for cell engineering. Li et al. [13] used PDA to modify PLT to form a stable coating on the membrane surface of PLT (Table 3). The nano/micromotor was driven by NIR light, and the payload, such as doxorubicin (DOX), was combined with PDA, or entered the PLT through endocytosis and was released in a slightly acidic environment, which normally occurs in the tumor microenvironment. In the MCF7 breast cancer tumor-bearing mouse model, it was proved that the vein-injected nano/micromotors penetrated deeply into solid tumors with NIR irradiation (1.9- and 2.9-fold higher than the group without NIR and the DOX group) and had the largest cell necrosis area, reaching about 23.7%. Other hemocytes also play an important role in nano/micromotor therapy. Munerati et al. [71] reported the drug-encapsulating ability of macrophages. Shao et al. [72] developed a hybrid neutrophil nano/micromotor system, which was guided along the chemoattractant gradients secreted by Escherichia coli for targeted drug transport in vitro. Despite the application of blood cells in the delivery system, for other nano/micromotors, they may be inevitable barriers to targeting certain positions. To avoid the attachment of blood cells, Li et al. [73] demonstrated an autonomous navigation system for nano/micromotors with high-performance path planning in different environments and realistic settings, enabling the recognition of red blood cells and cancer cells.

Nano/micromotors are also associated with thrombosis in the blood vessel. Doctors use various other materials to block blood vessels for treating tumors, fistulas, and arteriovenous malformations [83]. However, the embolization of these materials has the possibility of severe complications, such as stroke, due to the low selectivity and unintentional blockage of the nontargeted sites [84]. Nano/micromotors have been explored to achieve selective embolization. Law et al. [85] designed a superparamagnetic particle coated with thrombin (Figure 3b). They deduced that the motor would agglutinate in a certain magnetic field. A stronger magnetic field was applied to the target site, while the other areas had weaker magnetic fields that could not propel the motors together, avoiding side effects. Subsequent experiments showed that this nano/micromotor achieved the expected effect in microfluidic channels and porcine organs. Although embolization could cure diseases, one needs to be extremely careful when an embolism occurs in the body. Atherosclerosis is the main cause of embolism. The formation of atherosclerosis involves a variety of adverse factors [86]. There are many problems in atherosclerosis therapy, such as single drugs, lack of targeting, and severe side effects, which might be solved by the application of nano/micromotors [87,88]. Li et al. [74] propose a NIR light-driven multifunctional mesoporous/macroporous tubular nano/micromotor that could rapidly target damaged blood vessels and release different drugs (Table 3). Its motility effect could enable it to penetrate the plaque site, and the thermal effect caused by NIR irradiation could also achieve photothermal ablation of inflammatory macrophages. Moreover, these nano/micromotors are loaded with vascular endothelial growth factor (VEGF) in macropores and paclitaxel (PTX) in mesopores. Due to different loading apertures, VEGF releases rapidly, and by contrast, PTX releases slowly, to achieve synergistic treatment of atherosclerosis. The results in the mouse model showed that the nano/micromotor had a good therapeutic effect on atherosclerosis. Compared with the saline group (21.3%), the aortic lesion area could be reduced to around 3.1%. The same mesoporous/macroporous loading and sequential release of different drugs have also been used in thrombolytic therapy. Wan et al. [75] developed a mesoporous/macroporous silica/platinum nano/micromotor with PLT membrane (PM) modification (MMNM/PM) (Table 3). Regulated by the special proteins on the PM, the motor targeted the thrombus site, and then the PM would be ruptured under NIR irradiation to achieve ideal sequential drug release, including the rapid release of thrombolytic urokinase (UK) (3 h). and slow-release anticoagulant heparin (Hep) (>20 days). It has shown a good curative effect in the thrombosis mouse model. When the treatment time was 7 days, the relative volume of the thrombus decreased to less than 0.05. The PM-coated structure reduced the leakage of drugs before reaching the designated site, which decreased the dosage of drugs and improve the effect. Huang et al. [76] also used the PLT membrane to design a similar NIR-driven Janus nanomotor for the treatment of vascular plates (Table 3). Visible light in the spectrum can also be used to drive nano/micromotors. Removing toxins from the blood is not the same as in the stomach. The components in the blood hamper the absorption of toxic substances. Pacheco et al. [77] have demonstrated the successful operation of light-driven nano/micromotors in diluted blood for toxins removal (Table 3). The polymeric PLGA layer in the nano/micromotor imparted it with hydrophilicity and anti-biofouling properties to prevent proteins and RBCs adhesion and subsequent pore blocking of the nano/micromotors. However, it required a relatively high glucose level (0.22 M) for propulsion compared with physiological levels.

As an important signaling molecule, nitric oxide (NO) plays an important role in cardiovascular diseases and has been applied in the treatment of cardiovascular-related diseases for many years [89]. The NO can be the power source of the nano/micromotors. Tao et al. [78] designed NO-driven silica nanomotors with bowl-shaped mesoporous structures targeted to the thrombus surface through the modification of arginine-glycine-aspartate (RGD) polypeptides and simultaneous loading with L-arginine (LA) and the thrombolytic drug UK (Figure 3c, Table 3). LA reacted with excess ROS in the thrombus microenvironment to generate NO, thereby propelling the movement of nanomotors. The researchers integrated multiple functions. The reaction not only improved the retention efficiency and utilization of drugs at the thrombus site but also eliminated ROS and reduced oxidative stress in inflammatory endothelial cells. However, only converting chemical fuel or external energy into mechanical motion is not powerful enough to propel the nano/micromotors. The hybrid nano/micromotor, mixing two propulsion mechanisms, can combine the advantages of both to improve drug delivery efficiency and therapeutic effect. Wu et al. [79] constructed a nanomotor with bi-propulsion by utilizing the covalent binding and self-assembly of β-cyclodextrin(β-CD) and L-arginine (LA) immobilized with Au nanoparticles (Table 3). It used an anti-VCAM-1 monoclonal antibody (aV) to target damaged blood vessels, and NIR laser irradiation, as a driving force, ablated inflammatory macrophages through a photothermal effect. NO released by nanomotors could be used as another power and therapeutic agent to promote endothelial repair at the plaque site. LA played a role in eliminating ROS and β-CD promoted the removal of cholesterol from foam cells. Fang et al. [80] designed and fabricated PDA nanomotors (PDANMs) modified with RGD peptide and loaded with the UK (Table 3). PDA substrate had a good photothermal conversion effect. The guanidine group of L-arginine could interact with ROS in the thrombus microenvironment to generate nitric oxide. The emphasis of these dual-drive nano/micromotors is different, exploring various aspects of the delivery system in the blood vessels, which shows a good prospect of medical application.

Among cardiovascular diseases, acute ischemic stroke has been the leading cause of long-term disability [90]. Nano-/micromotors powered by a magnetic field or sonodynamic force is helpful with current therapeutic approaches. Tissue plasminogen activator (tPA) is the only thrombolytic drug that FDA has approved for hemorrhagic stroke patients for more than a decade [81,91]. Hu et al. [81] developed a novel nano/micromotor incorporating tPA into porous magnetic iron oxide (Fe_3_O_4_)-microrods (tPA-MRs) for targeted dissolution of the ischemic stroke thrombosis in the distal middle cerebral artery (Table 3). They found that intra-arterial injection of tPA-MRs could target brain blood clots in vivo under the guidance of an external magnet, where tPA was subsequently released at the site of embolization. In a mouse stroke model, injection of tPA-MRs had a much lower concentration of 0.13 mg/kg than tPA injection alone (10 mg/kg), and it took less than 1/3 of the time to lyse the blood clot. However, magnetically powered carriers have a number of limitations in a complex operation. Flexibility and controllability are crucial for nano/micromotors in achieve their applications. Yu et al. [92] designed a magnetic trimer-like nano/micromotor composed of three magnetic Janus colloids with different diameters, manifesting actively controlled manipulation. It could pass through a complex simulated vascular channel. Moreover, Cao et al. [82] constructed Janus rod (JR)-shaped nano/micromotors propelled by the ejection of O2 bubbles and the cavitation effect produced by the ultrasonication of O_2_ (Figure 3d, Table 3). The nano/micromotor could achieve sonodynamic therapy (SDT) of thrombolysis without any other thrombolytic drugs. The lower limb thrombosis model showed 3.4-fold higher accumulations at the clot site, and ultrasonication further increased the thrombus retention by 2.1 times. Interestingly, bio-hybrid nano/micromotors are also designed to treat thrombosis. It was reported that bacteria-templated nano/micromotors [93] or directly bacteria-propelled motors [94] had been designed to promote efficient penetration and thrombolytic efficacy.
Figure 3Nano/micromotors in the blood vessels. (**a**) Fe_3_O_4_ nanoparticle-loaded RBC motors could be propelled by ultrasound fields. Reprinted with permission from ref. [69]. Copyright © 2022, American Chemical Society. (**b**) Superparamagnetic motors coated with thrombin could agglutinate in a certain magnetic field and block the vessel. Reprinted with permission from ref. [76]. (**c**) The LA of NO-driven silica motors with bowl-shaped mesoporous could react with ROS to produce NO to propel the motor. Reprinted with permission from ref. [78,85]. (**d**) The ejection of O_2_ bubbles and the cavitation effect produced by the ultrasonication of O_2_ could be the power source of the Janus rod (JR)-shaped motors. Reprinted with permission from ref. [82,92].
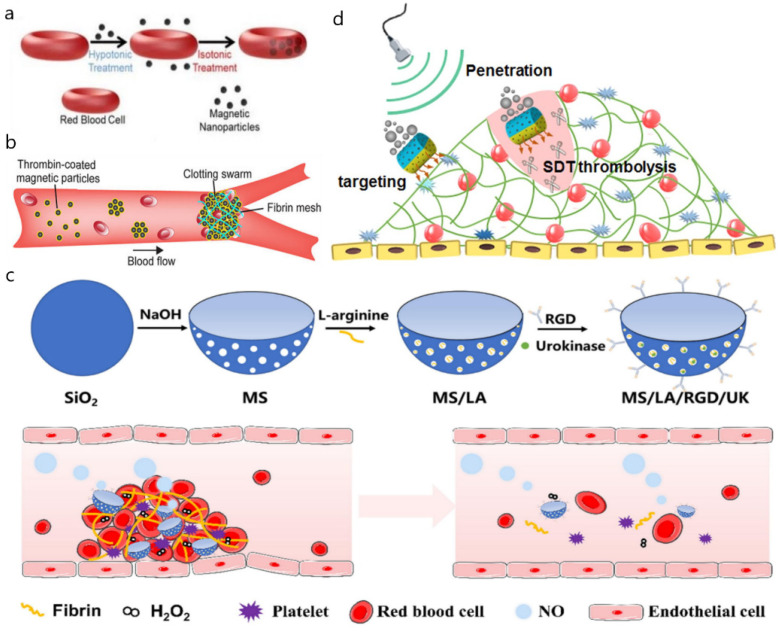


Current problems and remaining studies: (A) Different blood vessels have different blood velocities, so strong propulsion does not apply to all blood vessels. In the future, more flexible and adjustable micromotors will be needed for variable blood vessels (such as capillaries [95]). We may refer to the streamlined structure of fish in the design of micromotors. (B) The real-time imaging of micromotors in blood vessels has become an urgent problem to be solved [96]. We suggest that future works may find a strong signal source, such as the magnetic microswarm for MRI contrast enhancer [97], which is harmless to the human body. It could be a part of the micromotor, and then through the instrument in vitro, it could provide imaging for the precise operation of micromotors. (C) The micromotor designed by Li et al. [74] could rapidly target damaged blood vessels. The vessels in different normal tissues, such as hepatic sinusoids and glomeruli, could also be a target. We suggest that future works could add specific recognition components to the micromotor to target liver and kidney tissues and realize the monitoring of liver and kidney function in the future.

## 5. Nano/Micromotors in Other Lumens

There are also various other natural cavities in the human body, such as the vagina, uterus, etc. Compared with the above-mentioned lumen, they also have different microenvironments and properties, so the nano/micromotors designed for these natural lumens are also different.

The vaginal environment is complex and changeable. The composition and content of the vaginal endothelium, cervical mucus, and various secretions vary over time and with hormone levels [98]. Common vaginal administration routes are vaginal tablet, gel, and film dosage forms, which have the advantages of precise drug delivery, better drug stability, and low cost [99]. Although many nanomaterials have been used for vaginal delivery, there seems to be no use for nano/micromotors [100]. However, the sperm-centered bio-hybrid nano/micromotor has a good biomedical application in the female reproductive tract. The natural function of sperm is to fertilize the oocyte in the human body. Medina-Sanchez et al. [101] reported the use of nano/micromotors to guide or transport motile and immotile sperms to eventually overcome two of the main male infertility problems: oligozoospermia (low sperm count) and asthenozoospermia (poor sperm motility), respectively. Sperm cells also are good carriers compared to such conventional carriers as liposomes, nanoparticles, etc. [102], for the treatment of female reproductive tract cancers. The same team [102] used bovine sperm to load the drug and deliver it to HeLa cells. Compared with the DOX solution alone, sperm cells loaded with DOX had a higher killing rate of Hela cells (87% > 55%). For more precise drug delivery, the team designed a special quadruped structure with a tubular body and four arms (Table 4). The sperm head could enter the tubular body and release the sperm when it encountered obstacles or cell bodies, performing accurate kills. However, the use of bovine sperm may cause severe immune responses and inflammation [103], which limits the application of bovine sperm. The significant advantages of human sperm nano/micromotors cannot be ignored; they include higher payload, fewer immune reactions, avoidance of dose dumping, self-propulsion, and the delivery of hydrophilic drugs, such as DOX, daunorubicin, etc. Sperm cells possess the somatic cell-fusion ability to help in improving the transfer and availability of the actives at the targeted site without accumulation in the healthy tissue [104]. Meanwhile, biohybrid nano/micromotors not only have to be guided toward a target, but they also have to be released close to this target in a controlled manner [105]. Studies have shown that sperm motors can effectively travel in bovine tubal fluid or artificial tubal fluid of various viscosities under the guidance of a magnetic field [106]. Motors composed of sperm can even fight against continuous or pulsating blood flow [107] (Figure 4, Table 4). However, the assembly of the nano/micromotor affects the swimming of sperm and slows down the traveling speed of the sperm nano/micromotor, so we need a more reasonable structure and proper magnetic guidance to reduce this side effect [108].

In addition to being applied in the female reproductive tract, nano/micromotors have also shown great potential in some other lumens. Zhang et al. [109] demonstrated a bioinspired, magnetically-driven micromotor consisting of nanoparticle-modified algae for active therapeutic delivery to treat lung disease. Drug delivery with the help of microrobots has the potential for use in targeted therapy in the eye. Wu et al. [110] reported a kind of micromotor that could penetrate the vitreous humor and reach the retina. After being injected into the vitreous humor of the eyes, micromotors are magnetically driven toward the retina, and this progress could be observed by optical coherence tomography (OCT). However, the use of microrobots has the limitation of leaving magnetic nanoparticles (MNPs) in the eye, which can lead to side effects. Kim et al. [111] proposed a bilayer hydrogel microrobot capable of recycling MNPs after drug delivery, overcoming the limitations of existing microrobots. The bilayer hydrogel microrobot consists of an MNP layer and a therapeutic layer. When an alternating magnetic field (AMF) is applied at the target point, the therapeutic layer dissolves to deliver drug particles, and the MNP layer can then be retrieved using the magnetic field. In vitro bovine vitreous and cell tests demonstrated the vitreous migration potential of the nano/micromotor and therapeutic efficacy against retinoblastoma Y79 cancer cells.

Current problems and remaining studies: (A) Sperm micromotors derived from other species may cause severe immune responses in the human body, and the raw materials for human sperm micromotors are difficult to obtain. We suggest that future works should design sperm-like artificial micromotors to reduce cost and avoid immune reactions.

## 6. Summary and Expectation

We reviewed the application of nano/micromotors in biological lumens, especially in drug delivery. Nano/micromotors can actively locomote to the aimed sites, in contrast with passive drug carriers. With self-propulsion or external propulsion forces, these motors can localize in a certain area and the cargo, such as drugs, can release in this area, improving the therapeutic effect and reducing the side effects. Nano/micromotors are considered to have similar effects in various biological lumens, such as the central spinal canal, bronchial, brain ventricle, marrow cavity, etc. However, motor-induced immune response in the complex environments of lumens might limit the application of nano/micromotors. The components of the nano/micromotors and their degradation should be optimized to reduce side effects in the human body. With continuous research efforts to overcome the roadblocks, nano/micromotors will soon find wide application.

An example of the advances achieved in overcoming some of these challenges is a homopolymer poly[6-(4-methoxy-4′-oxyazobenzene) hexyl methacrylate] (PAzoMA) particle with Pt half-coating on the surface [112]. This Janus self-propelled motor can change its shape precisely (from spherical to elliptical) at a brief optical signal (450 nm) and maintain this deformation until the next signal reception. Compared with the spherical shape, the motion speed of the deformed elliptical shape is relatively low. Such deformable self-propelled motors have the potential to optimize drug diffusion efficiency, enabling velocity changes and long-term monitoring in diseased regions. The speed control of the self-propelled motor could also be adjusted by temperature control [113]. Although all these problems cannot be solved immediately, we expect that the study of self-propelled motors will gradually grow and become a productive platform for clinical treatments.

## Figures and Tables

**Figure 4 micromachines-13-01780-f004:**
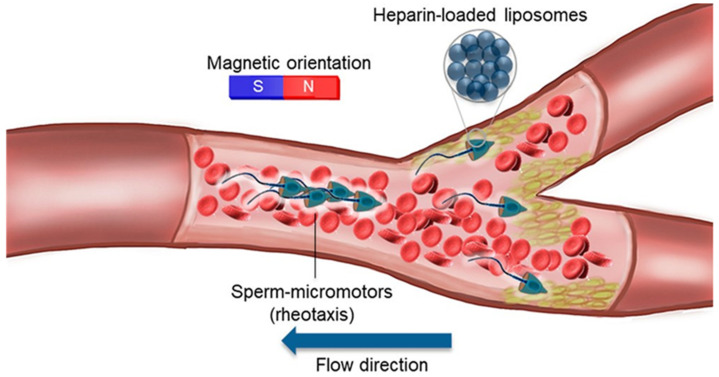
Nano/micromotors in other lumens. The motors composed of sperm can even fight against continuous or pulsating blood flow. Reprinted with permission from ref. [107]. Copyright © 2022 American Chemical Society.

**Table 1 micromachines-13-01780-t001:** A summary of micro/nanomotor size, power source, locomotion speed, lifespan, biocompatibility, and applicable environment in the gastrointestinal tract.

Name	Size	Power Source	Locomotion Speed	Lifespan	Biocompatibility	Applicable Environment
Double-conical Zinc Micromotors [31]	20 μm	Chemical power: Zn	110 μm/s	15 s	/	Stomach
PEDOT/Zn Bilayer Micromotors [32]	20 μm	Chemical power: Zn	~60 μm/s	~10 min	showed no apparent toxicity	Stomach
Multicompartment Tubular Micromotors [33]	~15 μm	Chemical power: Zn	~70 μm/s	15 s–60 s (the length of the Zn segment)	showed no apparent toxicity	Stomach and Duodenum
Mg-based Micromotors [34]	~20 μm	Chemical power: Mg	~120 μm/s	~6 min	showed no apparent toxicity	Stomach
Bubble-propelled Janus Ga/Zn Micromotors [35]	~8.9 μm	Chemical power: Zn	383–161.2 μm/s	0.55–5.2 min	showed no apparent toxicity	Stomach
Motor Toxoids [36]	~25 μm	Chemical power: Mg	160–200 μm/s	/	showed no apparent toxicity	Stomach and Small intestine
Suction-cup-inspired Micromotors [37]	~300 μm	Chemical power: Mg	200 μm/s	~16 s	/	Stomach
Biological Chemotaxis-guided Self-thermophoretic Micromotors [38]	~80 μm	Photocatalytic activity	2 μm/s	/	/	Colorectum

/: The data are not specified in the articles.

**Table 2 micromachines-13-01780-t002:** A summary of micro/nanomotor size, power source, locomotion speed, lifespan, biocompatibility, and applicable environment in the urinary tract.

Name	Size	Power Source	Locomotion Speed	Lifespan	Biocompatibility	Applicable Environment
Active Hybrid Microcapsule Motors [54]	~2.32 μm	Powered by the biocatalytic decomposition of urea	~11 μm/s	over 10 min	/	Bladder
Enzyme-powered Polymer Micromotors [55]	~1 μm	Powered by the biocatalytic decomposition of urea	10.67 μm/s	/	showed no apparent toxicity	Bladder
Enzyme-powered Janus Platelet Micromotors [57]	~2 μm	Powered by the biocatalytic decomposition of urea	~6 μm/s	over 30 min	/	Bladder
Enzyme-photocatalyst Tandem Micromotors [60]	~15 μm	Enzyme and the photocatalytic activity	~3.3 μm/s	over 2 h	/	Bladder/Urinary catheters
Photothermal Interference (PTI) Urease-modified Polydopamine (PDA) Micromotors [61]	~330 nm	Enzyme and the photocatalytic activity	increased mean square displacement	/	showed no apparent toxicity	Bladder

/: The data are not specified in the articles.

**Table 3 micromachines-13-01780-t003:** A summary of micro/nanomotor size, power source, locomotion speed, lifespan, biocompatibility, and applicable environment in the blood vessels.

Name	Size	Power Source	Locomotion Speed	Lifespan	Biocompatibility	Applicable Environment
RBC Micromotors [69]	~7 μm	Ultrasound propulsion and magnetic guidance	5 μm/s	/	anti-phagocytosis capability against macrophages	Blood vessel
Red Blood Cell-mimicking (RBCM) Micromotors [70]	~2.1 μm	Ultrasound propulsion and magnetic guidance	~7 μm/s	/	showed no apparent toxicity	Blood vessel
Engineered PLT Micromotors [13]	~2 μm	Photocatalytic activity	2.2~12.2 μm/s	/	showed no apparent toxicity	Blood vessel
NIR Light-driven Multifunctional Mesoporous/Macroporous Tubular Micromotors [74]	~8 μm	Photocatalytic activity	~5.4 μm/s	/	showed no apparent toxicity	Blood vessel
Platelet-derived Porous Micromotors [75]	410 nm	Photocatalytic activity	increased mean square displacement	/	showed no apparent toxicity	Blood vessel
Platelet-derived Nanomotor Coated Balloon [76]	~500 nm	Photocatalytic activity	increased mean square displacement	/	showed no apparent toxicity	Blood vessel
Functional Coatings Enable Navigation of Light-propelled Micromotors [77]	~20 μm	Photocatalytic activity	~8.2 μm/s	/	showed no apparent toxicity	Blood vessel
Nitric Oxide-driven Nanomotors [78]	~255 nm	Chemical power: NO	~3.5 μm/s	/	showed no apparent toxicity	Blood vessel
β-cyclodextrin/L-arginine/Au Nanomotors with Dual-mode Propulsion [79]	~250 nm	Chemical power: NO and Photocatalytic activity	5~8 μm/s	/	showed no apparent toxicity	Blood vessel
Dual Drive Mode Polydopamine Nanomotors [80]	~200 μm	Chemical power: NO and Photocatalytic activity	~2.32 μm/s	/	showed no apparent toxicity	Blood vessel
tPA-porous Magnetic Microrods [81]	~1.3 μm	Magnetic guidance	/	/	showed no apparent toxicity	Blood vessel
Ultrasound-propelled Janus Rod-shaped Micromotors [82]	~1.3 μm	Ultrasound propulsion	penetration in agarose gels	/	showed no apparent toxicity	Blood vessel

/: The data are not specified in the articles.

**Table 4 micromachines-13-01780-t004:** A summary of micro/nanomotor size, power source, locomotion speed, lifespan, biocompatibility, and applicable environment in other lumens.

Name	Size	Power Source	Locomotion Speed	Lifespan	Biocompatibility	Applicable Environment
Sperm-hybrid Micromotors [102]	>50 μm	Sperm propulsion and Magnetic guidance	~50 μm/s	/	/	Female reproductive tract
Sperm-driven Micromotors [107]	>50 μm	Sperm propulsion and Magnetic guidance	~76 μm/s	/	showed no apparent toxicity	Blood vessel

/: The data are not specified in the articles.

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
