# Peer review of "Applications of Nano/Micromotors for Treatment and Diagnosis in Biological Lumens"

_micromachines, 2022, doi:10.3390/mi13101780_

Round 1
Reviewer 1 Report
In this manuscript Huang et al. summarize recent advances in the application of micro/nano motors in biological lumens. They comprehensively cover the application in GI tract (primarily of motors using Zn and Mg as active components, the urinary tract with primarily enzyme based motors, and blood vessels where they focus on light activated and iron oxide based motors. They finally provide a brief summary of applications with bio-hybrid motors in female reproductive tract.
For the community specifically interested in biological lumens and micro-nano motors I think this review will be a certainly useful and is a welcome addition alongside numerous reviews covering biological applications of micro-nano motors generally. This review is well organized, and the figures are largely well selected. I would recommend additional proof-reading of the text, especially the introduction. I have some minor comments that could be helpful:
1. The authors should clearly distinguish when the studies were performed in-vivo and in lumen-like simulated environments. Sometimes this distinction is unclear.
2. I appreciate that the authors list current problems for each section, but the solutions that the authors suggest are largely speculative and need to be substantiated with supporting evidence of what makes these feasible and citations.
3. Towards the end of Section 3 (line 232) the authors seem to suggest the use of electrically driven motors in urinary tracts. How do they envision the implementation of such a system in-vivo?
4. There are works in the community of micro/nano motors that show behavior of these systems in fluid flows (rheotaxis, cross-stream migration) and in complex environments such as mucus. Even though these are not directly performed in biological lumens the results might be relevant and worth mentioning.
5. the following citations might be relevant:
Wu, Zhiguang, Jonas Troll, Hyeon-Ho Jeong, Qiang Wei, Marius Stang, Focke Ziemssen, Zegao Wang et al. "A swarm of slippery micropropellers penetrates the vitreous body of the eye." Science advances 4, no. 11 (2018): eaat4388.
Walker, Debora, Benjamin T. Käsdorf, Hyeon-Ho Jeong, Oliver Lieleg, and Peer Fischer. "Enzymatically active biomimetic micropropellers for the penetration of mucin gels." Science Advances 1, no. 11 (2015): e1500501.
Zhang, F., Zhuang, J., Li, Z. et al. Nanoparticle-modified microrobots for in vivo antibiotic delivery to treat acute bacterial pneumonia. Nat. Mater. (2022). https://doi.org/10.1038/s41563-022-01360-9
Author Response
- The authors should clearly distinguish when the studies were performed in-vivo and in lumen-like simulated environments. Sometimes this distinction is unclear.
Response: Thank you for your valuable suggestion. We have cleared the distinction when the studies were performed in-vivo and in lumen-like simulated environments. (line200, line230, line240, line271)
- I appreciate that the authors list current problems for each section, but the solutions that the authors suggest are largely speculative and need to be substantiated with supporting evidence of what makes these feasible and citations.
Response: Thank you for your valuable suggestion. We added some supporting evidence of the current problems for some sections. (line250-251, line404-405, line407-408)
- Towards the end of Section 3 (line 232) the authors seem to suggest the use of electrically driven motors in urinary tracts. How do they envision the implementation of such a system in-vivo?
Response: Thank you for your valuable suggestion. The answer to the question has been added in section (line247-249).
4&5. There are works in the community of micro/nanomotors that show the behavior of these systems in fluid flows (rheotaxis, cross-stream migration) and in complex environments such as mucus. Even though these are not directly performed in biological lumens the results might be relevant and worth mentioning. The following citations might be relevant: ……
Response: Thank you for your valuable suggestion. We have added the relevant citations that are attached to the suggestion in suitable places. (line120-125, line454-456, line457-460)
Reviewer 2 Report
The manuscript provided an overview of recent progress and challenges in developing new treatment and diagnostic approaches inside biological lumens via micro/nanomotors. This is an interesting topic about a fast-developing area, where such review would be very useful to the community. The authors well summarized research works in the past decades about both the operational principles of micro/nanomotors and their biomedical applications. I found the manuscript contains information that are quite explanatory for readers from very different disciplines. However, some revisions are needed to improve its quality. The authors should address the comments, questions and suggestions listed below:
1. This review paper is organized by discussing the micro/nanomotors working in various lumens. However, there lacks a high-level and general discussion about what are the specific challenges of deploying micro/nanomotors in bio lumens. For instance, I suggest the authors to expand the discussion in the last paragraph of the introduction section, and remake a new schematic for figure 1.
2. Following comment 1, it would be helpful if authors first discuss the features of all kinds of micro/nanomotors that are capable of doing locomotion and can be remotely controlled in lumens. A summary of advantages and limits of those micro/nanomotors would help the readers to better understand why they are chosen for specific applications.
3. A table summarizing micro/nanomotors’ size, working principle, locomotion speed, fuel, biocompatibility, applicable environment would be very useful in addition to the figures.
Author Response
- This review paper is organized by discussing the micro/nanomotors working in various lumens. However, there lacks a high-level and general discussion about what are the specific challenges of deploying micro/nanomotors in bio lumens. For instance, I suggest the authors expand the discussion in the last paragraph of the introduction section and remake a new schematic for figure 1.
Response: Thank you for your valuable suggestion. We have made a high-level and general discussion about what are the specific challenges of deploying micro/nanomotors in bio lumens in the last paragraph of the introduction section. (line50-54)
- Following comment 1, it would be helpful if authors first discuss the features of all kinds of micro/nanomotors that are capable of doing locomotion and can be remotely controlled in lumens. A summary of the advantages and limits of those micro/nanomotors would help the readers to better understand why they are chosen for specific applications.
Response: Thank you for your valuable suggestion. And we have discussed the features of all kinds of micro/nanomotors after the general discussion. (line54-58)
- A table summarizing micro/nanomotors’ size, working principle, locomotion speed, fuel, biocompatibility, and applicable environment would be very useful in addition to the figures.
Response: Thank you for your valuable suggestion. We made a table of micro/nanomotors’ size, locomotion speed, fuel, biocompatibility, applicable environment et al. (line564-566)
Reviewer 3 Report
The authors summarize the recent progresses of micro/nanomotors for biological applications. There are two comments that should be addressed:
1. There are many errors such as spelling and formatting in the manuscript.
2. The challenges of micro/nanomotors in biological applications need to be further discussed.
Author Response
- There are many errors such as spelling and formatting in the manuscript.
Response: We have corrected the spelling and formatting errors in the manuscript.
- The challenges of micro/nanomotors in biological applications need to be further discussed.
Response: We added the discussion in the last paragraph of the introduction. (line50-54)